# Return-Aligned Decision Transformer

**Tsunehiko Tanaka**                                          *tsunehiko@fuji.waseda.jp*
*Waseda University*

**Kenshi Abe**                                          *abekenshi1224@gmail.com*
*CyberAgent*

**Kaito Ariu**                                          *kaito_ariu@cyberagent.co.jp*
*CyberAgent*

**Tetsuro Morimura**                                          *morimura_tetsuro@cyberagent.co.jp*
*CyberAgent*

**Edgar Simo-Serra**                                          *ess@waseda.jp*
*Waseda University*

**Reviewed on OpenReview:** *https://openreview.net/forum?id=lTt2cTW8h1*

## Abstract

Traditional approaches in offline reinforcement learning aim to learn the optimal policy that maximizes the cumulative reward, also known as return. It is increasingly important to adjust the performance of AI agents to meet human requirements, for example, in applications like video games and education tools. Decision Transformer (DT) optimizes a policy that generates actions conditioned on the target return through supervised learning and includes a mechanism to control the agent's performance using the target return. However, the action generation is hardly influenced by the target return because DT's self-attention allocates scarce attention scores to the return tokens. In this paper, we propose Return-Aligned Decision Transformer (RADT), designed to more effectively align the actual return with the target return. RADT leverages features extracted by paying attention solely to the return, enabling action generation to consistently depend on the target return. Extensive experiments show that RADT significantly reduces the discrepancies between the actual return and the target return compared to DT-based methods.

## 1 Introduction

Offline reinforcement learning (RL) focuses on learning policies from trajectories collected in offline datasets (Levine et al., 2020; Fujimoto & Gu, 2021; Yu et al., 2021; Jin et al., 2021; Xu et al., 2022). While many offline RL methods aim to optimize policies for maximum cumulative rewards (returns), they typically produce a single policy fixed to a specific performance level. This rigidity is problematic in scenarios requiring agents with varying skill levels, as it forces developers to iteratively adjust reward functions and retrain models for each desired performance. Obtaining lower-performing policies from intermediate model checkpoints during training is also unsuitable for flexible performance adjustments. This is because accurately assessing their performance requires online evaluation, which is often difficult to conduct in practice. These limitations make it difficult to tailor policies to specific performance requirements.

To overcome these limitations, we propose training a single model that can achieve any desired target return. By simply adjusting the *target return* parameter, one can obtain agents spanning a wide range of performance levels. This capability streamlines the policy development process in real-world applications, where heterogeneous performance is often required:

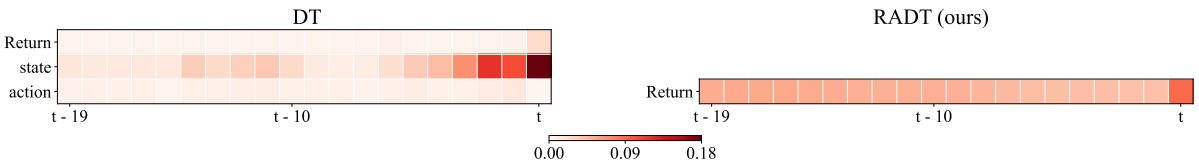

Figure 1: **Comparison of attention scores between DT and RADT trained on the MuJoCo ant-medium-replay dataset.** The figure shows the attention scores assigned to input sequences by the first self-attention layer in DT and the first SeqRA in RADT. Each value represents the episode average of the attention scores. DT assigns minimal attention to return-to-go tokens, while RADT assigns attention exclusively to return-to-go tokens, thereby improving return alignment in action generation.

**Example 1** (Creating AI opponents in video games and educational tools)**.** Consider implementing AI opponents using RL in video games (Yannakakis & Togelius, 2018) and educational tools (Singla et al., 2021) development. Developers implement AI opponents with varied skill levels to ensure players, from beginners to experts, are appropriately challenged (Shen et al., 2021; Wu et al., 2023a). Our proposed approach allows developers to interactively adjust the target return to produce a diverse set of AI opponents with different skill levels. This significantly reduces development overhead, enhancing game quality by providing tailored experiences for each player.

**Example 2** (Human motion generation)**.** Human motion generation in robotics and animation (Zhu et al., 2024) often uses RL to produce natural and controllable human pose sequences (Jiang et al., 2024; Peng et al., 2022). Our approach enhances controllability by enabling a single policy to adjust its behavior based on the target return. For example, using forward velocity as a reward function, a single policy model can generate walking motions for low target returns and running motions for high target returns. This flexibility improves the efficiency and versatility of motion synthesis for various downstream tasks (Rempe et al., 2023; Xie et al., 2024).

For these applications, Decision Transformer (DT) (Chen et al., 2021) is a promising method that enables control over an agent's performance via a target return. DT optimizes a policy through supervised learning to generate actions conditioned on the target return. Specifically, it takes a sequence comprising desired future returns (also known as return-to-go), past states, and actions as inputs, and outputs actions using a transformer architecture (Vaswani et al., 2017). In the self-attention mechanism of the transformer, each token selectively incorporates features from other tokens based on their relative importance, also known as the attention score. DT leverages the self-attention mechanism to propagate return-to-go tokens across the input sequence, enabling the conditioning of action generation on the target return. However, despite this conditioning, DT often obtains an actual return that diverges significantly from the target return. Our analysis, as illustrated in Fig. 1, reveals that the self-attention mechanism assigns only minimal attention to the return-to-go tokens, causing the return-to-go information to be nearly lost as it propagates through the network. This suggests that the return-to-go information nearly vanishes after passing through DT's self-attention, and leads DT to generate actions that are largely independent of the target return.

In this paper, we propose the Return-Aligned Decision Transformer (RADT), a novel architecture designed to align the actual return with the target return. Our key idea is to separate the return-to-go sequence from the state-action sequence so that the return-to-go sequence more directly influences action generation. To achieve this, we adopt two complementary design strategies: *Sequence Return Aligner* (SeqRA), described in Sec. 4.1, and *Stepwise Return Aligner* (StepRA), discussed in Sec. 4.2. SeqRA processes return-to-go tokens from multiple past timesteps to capture long-term dependencies. In contrast, StepRA links each state or action token directly to its corresponding return-to-go token at the same timestep, thereby capturing their stepwise relationship. By integrating these two approaches, RADT effectively leverages return-to-go information throughout the decision-making. In our experiments, RADT significantly reduces the absolute error between the actual and target returns, achieving reductions of 54.9% and 34.4% compared to DT in the MuJoCo (Todorov et al., 2012) and Atari (Bellemare et al., 2013) domains, respectively. Ablation studies demonstrate that each strategy independently contributes to improving return alignment, and their combination further enhances the model's ability to match the actual return to the target return.

In summary, our contributions are as follows:

1. We introduce RADT, a novel offline RL approach designed to align the actual return with the target return.

2. RADT employs a unique architectural design that treats the return-to-go sequences distinctly from state-action sequences, enabling the return-to-go information to guide action generation effectively.

3. We present empirical evidence that RADT surpasses existing DT-based models in return alignment.

## 2 Preliminary

We assume a finite horizon Markov Decision Process (MDP) with horizon $T$ as our environment, which can be described as $\mathcal{M} = \langle \mathcal{S}, \mathcal{A}, \mu, P, \mathcal{R} \rangle$, where $\mathcal{S}$ represents the state space; $\mathcal{A}$ represents the action space; $\mu \in \Delta(\mathcal{S})$ represents the initial state distribution; $P : \mathcal{S} \times \mathcal{A} \rightarrow \Delta(\mathcal{S})$ represents the transition probability distribution; and $\mathcal{R} : \mathcal{S} \times \mathcal{A} \rightarrow \mathbb{R}$ represents the reward function. The environment begins from an initial state $s_1$ sampled from a fixed distribution $\mu$. At each timestep $t \in [T]$, an agent takes an action $a_t \in \mathcal{A}$ in response to the state $s_t \in \mathcal{S}$, transitioning to the next state $s_{t+1} \in \mathcal{S}$ with the probability distribution $P(\cdot|s_t, a_t)$. Concurrently, the agent receives a reward $r_t = \mathcal{R}(s_t, a_t)$.

**Decision Transformer (DT) (Chen et al., 2021)** introduces the paradigm of transformers in the context of offline reinforcement learning. We consider a constant $R^{\text{target}}$, which represents the total desirable return obtained throughout an episode of length $T$. We refer to $R^{\text{target}}$ as the target return. At each timestep $t$ during inference, the desirable return to be obtained in the remaining steps is calculated as follows:

$$\hat{R}_t = R^{\text{target}} - \sum_{t'=1}^{t-1} r_{t'} \tag{1}$$

We refer to $\hat{R}_t$ as return-to-go. DT takes a sequence of return-to-go, past states, and actions as inputs, and outputs an action $a_t$. The input sequence of DT[1] is represented as

$$\tau = (\hat{R}_1, s_1, a_1, \hat{R}_2, s_2, a_2, ..., \hat{R}_t, s_t). \tag{2}$$

Raw inputs, referred to as tokens, are individually projected into the embedding dimension by separate learnable linear layers for returns-to-go, state, and action respectively, to generate token embeddings. Note that from this point onwards, we will denote tokens as $\hat{R}_i, s_i, a_i$. The tokens are processed using a transformer-based GPT model (Radford et al., 2018). The processed token $s_t$ is input into the prediction head to predict the action $a_t$. The model is trained using either cross-entropy or mean-squared error loss, calculated between the predicted action and the ground truth from the offline datasets [2].

The transformer (Vaswani et al., 2017) is an architecture designed for processing sequential data, including the attention mechanism, residual connection, and layer normalization. The attention mechanism processes three distinct inputs: the query, the key, and the value. This process involves weighting the value by the normalized dot product of the query and the key. The weight is also known as the attention score, and is calculated as follows:

$$\alpha_{ij} = \text{softmax}(\langle q_i, k_\ell \rangle_{\ell=1}^n)_j, \tag{3}$$

where $\alpha_{ij} = 0, \forall j > i$ denotes a causal mask and $n$ denotes the input length. The causal mask prohibits attention to subsequent tokens, rendering tokens in future timesteps ineffective for action prediction. The $i$-th output token of the attention mechanism is calculated as follows:

$$z_i = \sum_{j=1}^n \alpha_{ij} \cdot v_j, \text{ where } \sum_{j=0}^n \alpha_{ij} = 1 \text{ and } \alpha_{ij} \geq 0. \tag{4}$$

---

[1] For practicality, only the last $K$ timesteps are processed, rather than considering the full inputs.
[2] $R^{\text{target}}$ is the total return of the trajectory in the dataset during training.

DT uses self-attention, where query, key, and value are obtained by linearly different transformations of the input sequence,

$$q_i = \tau_i W^q, \quad k_i = \tau_i W^k, \quad v_i = \tau_i W^v, \tag{5}$$

Layer normalization standardizes the token features to stabilize learning. Residual connection avoids gradient vanishing by adding the input and output of attention layers or feed-forward layers. For further details on DT, refer to the original paper (Chen et al., 2021).

DT conditions action generation by employing the self-attention mechanism to disseminate return-to-go information throughout the input sequence. Despite its design, DT cannot align the actual return with the target return. To address this challenge, our goal is to minimize the following absolute error between the target return $R^{\text{target}}$ and the actual return $\sum_{t=1}^{T} r_t$ using a single model:

$$\mathbb{E}\left[\left\|R^{\text{target}} - \sum_{t=1}^{T} r_t\right\|\right]. \tag{6}$$

## 3 Return Alignment Difficulty in DT

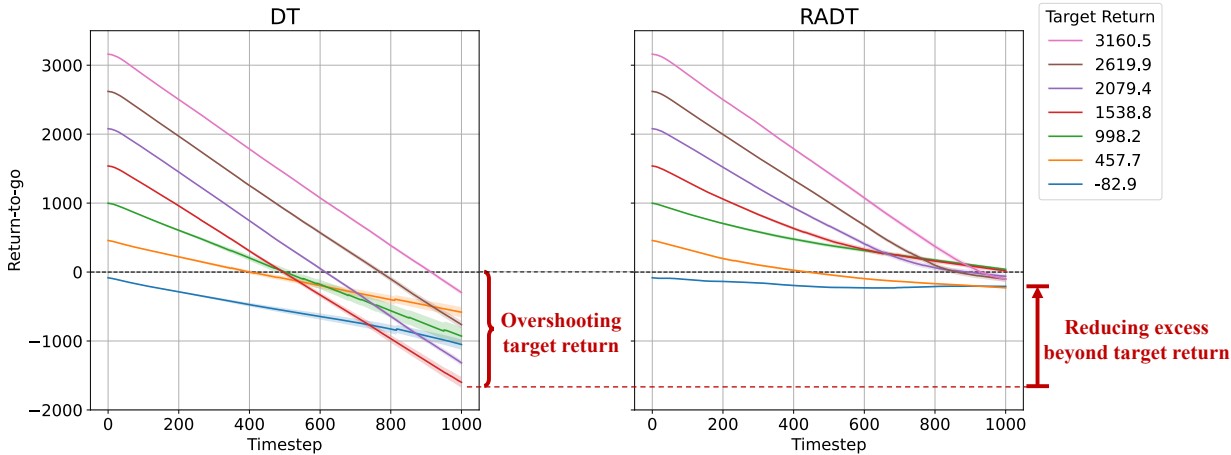

Figure 2: **Comparison of return-to-go transitions with varying target returns.** We evaluate DT and RADT on the ant-medium-replay dataset of MuJoCo. For each target return, we conduct 100 episodes, calculating the average and standard error of the target return at every step. The return-to-go of zero means that the actual return exactly matches the target return.

In order to understand the behavior of DT, we analyze how DT trained on the ant-medium-replay dataset of MuJoCo responds to the return-to-go throughout an episode. Ideally, the return-to-go should be zero at the end of the episode, which means that the actual return perfectly matches the target return. Figure 2 plots the changes in return-to-go input to the model during an episode. As shown in Fig. 2 (left), DT reaches values of return-to-go that fall significantly below zero for all target returns, and it obtains actual returns that differ greatly from the target returns.

We hypothesize that the problem lies in the architectural design of DT. Specifically, DT conditions action predictions on target returns through return-to-go tokens within the input sequence. These tokens compete for attention allocation in DT's self-attention mechanism, as shown in Eq. (4). When attention is predominantly given to state or action tokens, less attention is allocated to return-to-go tokens, weakening the influence of target returns on action prediction.

This hypothesis is supported by analyzing attention scores from DT's first self-attention layer, as shown in Fig. 1 (left). We observe that attention is strongly biased toward state tokens, with minimal attention to return-to-go tokens. This tendency could arise because the training data (Fu et al., 2020) includes samples generated by Markov policies that do not take return as input, causing DT to disregard return.

# 4   Return-Aligned Decision Transformer

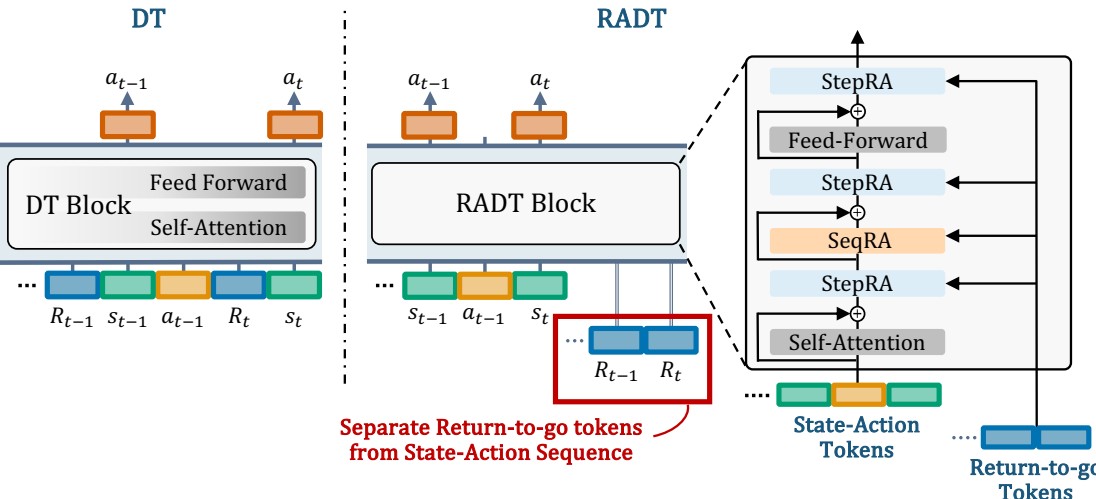

Figure 3: **Comparison between DT and the proposed RADT architecture**. DT processes a combined sequence of returns-to-go, states, and actions as input. In contrast, RADT separates the return-to-go from the state-action sequence and applies two Return Aligners: Sequence Return Aligner (SeqRA) and Stepwise Return Aligner (StepRA).

As previously discussed, DT struggles to align the actual return with the target return due to the under-allocation of attention scores to the return-to-go tokens. One intuitive way to solve this problem is to structure the transformer blocks such that the return-to-go tokens cannot be ignored when processing state and action tokens. To realize this intuition, we introduce *Return-Aligned Decision Transformer* (RADT).

We show the model structure of RADT in Fig. 3. We split the input sequence of $\tau$ in Eq. (2) into the return-to-go and other modalities: the return-to-go sequence $\tau_r$ and the state-action sequence $\tau_{sa}$

$$\tau_r = (\hat{R}_1, \hat{R}_2, ..., \hat{R}_t), \tag{7}$$

$$\tau_{sa} = (s_1, a_1, s_2, a_2, ..., s_t). \tag{8}$$

For practical purposes, RADT processes only the last $K$ timesteps of these sequences. In the transformer block, we first apply self-attention to the state-action sequence $\tau_{sa}$ to model dependencies within $\tau_{sa}$. We then apply our SeqRA and StepRA to $\tau_{sa}$ so that it strongly depends on the return-to-go sequence $\tau_r$. After the transformer blocks, the action $a_t$ is predicted from the $s_t$ token in the processed state-action sequence $\tau_{sa}$ by the prediction head. The model is trained using the cross-entropy or mean-squared error loss between the predicted action and the ground truth.

SeqRA and StepRA are designed as complementary alignment strategies that effectively condition the state-action sequence on the return-to-go tokens. The first strategy, *Sequence Return Aligner* (SeqRA), is illustrated in Fig. 4a. SeqRA can capture long-term dependencies involving distant return-to-go tokens by referencing multiple past timesteps in the return-to-go sequence. The second strategy, *Stepwise Return Aligner* (StepRA), is depicted in Fig. 4b. StepRA is designed to capture stepwise dependencies by having each state or action token always reference the corresponding return-to-go token of the same timestep. SeqRA enables the model to preserve the influence of past returns-to-go, while StepRA ensures an immediate response to the current return-to-go. These two strategies can be used independently or together. Guided by these strategies, the model naturally acquires a policy that approaches the target return at the end of an episode.

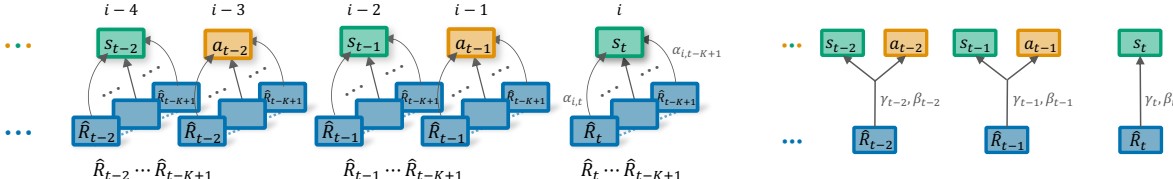

(a) **Sequence Return Aligner (SeqRA)**    (b) **Stepwise Return Aligner (StepRA)**

Figure 4: **Two strategies for conditioning the state-action sequence on the return-to-go sequence.** (a) In SeqRA, each state or action token is conditioned on all return-to-go tokens $(\hat{R}_{t-K+1}, \dots, \hat{R}_j)$, where $j \leq t$, to capture long-term dependencies. (b) In StepRA, each state or action token is conditioned only on the return-to-go token from the same timestep $\hat{R}_j$ to focus on stepwise relationships.

## 4.1 Sequence Return Aligner

SeqRA is designed to capture long-term dependencies within the return-to-go sequence $\tau_r$, enabling it to identify informative patterns that guide policy adjustments toward long-term objectives. This design leverages the sequence modeling capabilities introduced by DT in offline RL. We employ an attention mechanism to obtain the importance of each token in the return-to-go sequence and integrate the return-to-go sequence into the state-action sequence according to this importance. We make the state-action sequence $\tau_{sa}$ as the query, and the return-to-go sequence $\tau_r$ as both the key and value.

$$q_i = \tau_{sa,i} W^q, \quad k_j = \tau_{r,j} W^k, \quad v_j = \tau_{r,j} W^v. \tag{9}$$

These query, key, and value are applied to Eq. (3) to get the attention scores. The attention score $\alpha_{ij}$ represents how important the return-to-go token $\tau_{r,j}$ is compared to other return-to-go tokens in the return-to-go sequence. Note that we use a causal mask to ensure that tokens in the state-action sequence $\tau_{sa}$ cannot access future return-to-go tokens. By definition, $\sum_{j=t-K+1}^{t} \alpha_{ij} = 1$ and $\alpha_{ij} \geq 0$. As shown on the right side of Fig. 1, these attention scores are assigned exclusively to return-to-go tokens, ensuring that the model always references them without allocating any attention to state or action tokens. According to the attention scores, the return-to-go tokens are aggregated (see Fig. 4a),

$$z_i = \sum_{j=t-K+1}^{t} \alpha_{ij} \cdot \tau_{r,j} W^v, \tag{10}$$

The token $z_i$ serves as an embedding that captures the long-term dependencies of the return-to-go sequence in relation to $\tau_{sa,i}$.

We next incorporate the $z$ sequence into the state-action sequence $\tau_{sa}$. A simple addition may overemphasize $\tau_{sa}$, potentially preventing the effective use of the long-term dependencies of the return-to-go sequence from $z$. To address this, we learn parameters that adaptively adjust the scale of $z$ against $\tau_{sa}$ using a powerful method (Nguyen et al., 2022) from computer vision, which integrates two different types of features. The flow of this process is illustrated in Fig. 5. We concatenate $\tau_{sa,i}$ and $z_i$ as $[z_i; \tau_{sa,i}] \in \mathbb{R}^{2D}$ and obtain dimension-wise scaling parameters $\lambda$ through a learnable affine projection.

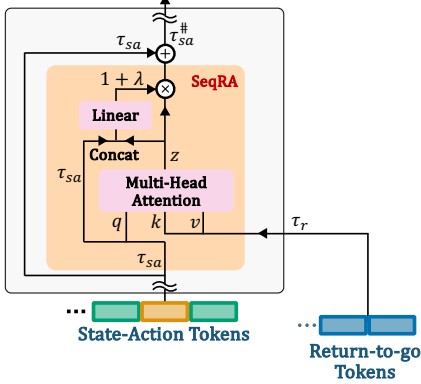

Figure 5: **Illustration of SeqRA.** We first apply an attention mechanism to obtain $z$ that integrates the return-to-go sequence $\tau_r$ into the state-action sequence $\tau_{sa}$. We then derive a scaling parameter $\lambda$ from $\tau_{sa}$ and $z$ to compute the weighted sum $\tau_{sa}^{\sharp}$.

The addition of $\tau_{sa,i}$ and $z_i$ is then formulated as follows, using $\lambda$,

$$\lambda_i = W[z_i; \tau_{sa,i}] + b, \tag{11}$$

$$\tau_{sa,i}^{\sharp} = (1 + \lambda_i) \otimes z_i + \tau_{sa,i}, \tag{12}$$

where $W \in \mathbb{R}^{D \times 2D}$ and $b \in \mathbb{R}^D$ are learnable parameters, and $\otimes$ denotes the Hadamard product. $\tau_{sa}^{\sharp}$ is the output of SeqRA and the input to the subsequent StepRA. By zero-initializing $W$ and $b$, the term $1 + \lambda_i$ allows the model to start with a baseline scaling factor of one (simple addition). At the start of training, we can also interpret this as a residual connection, a common technique in many transformer-based models (Vaswani et al., 2017; Chen et al., 2021; Dosovitskiy et al., 2021). As training progresses, $\lambda_i$ is updated, adjusting the balance between $z_i$ and $\tau_{sa,i}$.

## 4.2 Stepwise Return Aligner

StepRA associates the state or action token with the corresponding return-to-go token at the same timestep to capture their stepwise relationship. Specifically, at any timestep $j$, it applies a linear transformation to the state token $s_j$ or action token $a_j$ using weights inferred from the return-to-go token $r_j$. The linear transformation directly embeds the features of the return-to-go token into the state or action tokens. This process is applied separately and identically at each timestep. We train $\text{MLP}_{\gamma}$ and $\text{MLP}_{\beta}$ to predict the linear transformation weights $\gamma_j, \beta_j \in \mathbb{R}^D$. The process is formulated as follows:

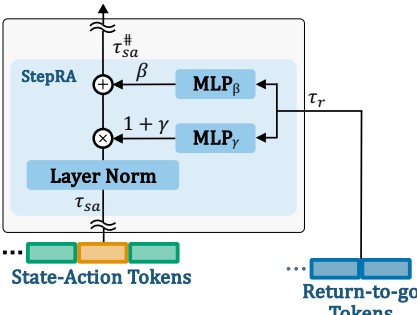

Figure 6: **Illustration of StepRA.** The state and action tokens are first normalized using layer normalization. They are then linearly transformed using the scaling factor $1 + \gamma$ and the shift parameter $\beta$, both inferred from the return token at the same time step.

$$s_j^{\sharp} = (1 + \gamma_j) \otimes \text{LayerNorm}(s_j) + \beta_j, \tag{13}$$

$$a_j^{\sharp} = (1 + \gamma_j) \otimes \text{LayerNorm}(a_j) + \beta_j, \tag{14}$$

$$\gamma_j = \text{MLP}_{\gamma}(r_j), \; \beta_j = \text{MLP}_{\beta}(r_j), \tag{15}$$

where $s_j$ represents the state token $\tau_{sa,2j-1}$, $a_j$ represents the action token $\tau_{sa,2j}$, and $r_j$ represents the return-to-go token $\tau_{r,j}$. The process flow of StepRA is illustrated in Fig. 6. We first perform layer normalization on the state and action tokens, and then apply this linear transformation. Because layer normalization ensures that the inputs share a uniform scale and distribution, the scaling and shifting parameters learned by the MLPs can work more stably. Similar to $1 + \alpha_i$ in Eq. (12), by zero-initializing the parameters of the linear layer, Eqs. (13) and (14) can be considered as the standard layer normalization at the beginning of training. The outputs $s^{\sharp}$ and $a^{\sharp}$ from the linear transformations are then concatenated to form the sequence $\tau_{sa}^{\sharp}$, which is used as input to the next components (self-attention, SeqRA, feed-forward).

## 4.3 Arrangement of SeqRA and StepRA

Figure 3 illustrates the placement of SeqRA and StepRA, which are inspired by the decoder of the original transformer (Vaswani et al., 2017). SeqRA corresponds to the cross-attention mechanism in the original transformer decoder, while StepRA serves as the layer normalization. SeqRA and cross-attention share a common function of integrating one type of information with another. In the case of SeqRA, it integrates the return-to-go sequence with the state-action sequence through an attention mechanism. For this reason, in RADT, we place SeqRA at the position of cross-attention in the original transformer (i.e., between self-attention and feed-forward layers). StepRA is designed to stabilize training by incorporating layer normalization. Therefore, we position StepRA immediately after each sublayer, including self-attention, SeqRA, and feed-forward.

## 5 Experiments

We conduct extensive experiments to evaluate our RADT's performance. First, we verify that RADT is effective in earning returns consistent with various given target returns, compared to baselines. Next, we demonstrate through an ablation study that the two types of return aligners constituting RADT are effective individually, and using both types together further improves performance. Furthermore, by comparing the transitions of return-to-go throughout an episode, we show that RADT can adaptively adjust its action in response to changes in return-to-go. Finally, we compare RADT with baselines in terms of maximizing actual return.

### 5.1 Datasets

We evaluate RADT on continuous (MuJoCo (Todorov et al., 2012)) and discrete (Atari (Bellemare et al., 2013)) control tasks in the same way as DT. MuJoCo requires fine-grained continuous control with dense rewards. We use four gym locomotion tasks from the widely-used D4RL (Fu et al., 2020) dataset: ant, hopper, halfcheetah, and walker2d. Atari requires long-term credit assignments to handle the delay between actions and their resulting rewards and involves high-dimensional visual observations. We use four tasks: Breakout, Pong, Qbert, and Seaquest. Similar to DT, we use 1% of all samples in the DQN-replay datasets as per Agarwal et al. (2020) for training.

### 5.2 Baselines and Settings

We utilize DT-based methods with various architecture designs as baselines, which enable action generation to be conditioned on the target return. Specifically, we use DT (Chen et al., 2021), StARformer (Shang et al., 2022), and Decision ConvFormer (DC) (Kim et al., 2024). For these baselines, we rely on their official PyTorch implementations. Further details about the baselines can be found in Appendix B.

In MuJoCo, for each method, we train three instances with different seeds, and each instance runs 100 episodes for each target return. In Atari, for each method, we train three instances with different seeds, and each instance runs 10 episodes for each target return.

Target returns are set by first identifying the range of cumulative reward in trajectories in the training dataset, specifically from the bottom 5% to the top 5%. This identified range is then equally divided into seven intervals, not based on percentiles, but by simply dividing the range into seven equal parts. Each of these parts represents a target return. Further details are provided in Appendix C.

### 5.3 Results

Figures 7 and 8 show the absolute error between the actual return and target return, plotted for each target return, for MuJoCo and Atari, respectively. We then average the absolute errors over all target returns, and present the mean and standard error of these averages across seeds in Tab. 1 for MuJoCo and Tab. 2 for Atari. In these figures and tables, the target returns, actual returns, and absolute errors are normalized with the largest target return set to 100 and the smallest set to 0. Overall, RADT significantly reduces the absolute error compared to DT, achieving 44.6% on MuJoCo and 65.5% on Atari, as calculated from Figures 9 and 11 in Appendix A.

**MuJoCo Domain.** In the MuJoCo domain, as shown in Tab. 1, RADT outperforms all baseline methods across all tasks. Furthermore, as illustrated in Fig. 7, RADT consistently achieves lower absolute error than the baselines in most target returns across all tasks. The baselines exhibit low sensitivity to changes in target returns and tend to bias their actual returns toward specific values. For example, significant errors are observed for baseline methods at lower target returns in the ant-medium-expert and walker2d-medium tasks. This is because the ant-medium-expert and walker2d-medium datasets contain few trajectories with low target returns. Additionally, StARformer achieves low absolute error for specific target returns (50 and 66.7) in the ant-medium environment in Fig. 7. However, its overall performance still results in a high absolute error, as shown in Tab 1. In contrast, RADT consistently shows lower absolute errors across a wide

Table 1: **Absolute error↓ comparison in the MuJoCo domain.** Target returns are split into seven equally spaced points from the bottom 5% to the top 5% of the dataset. We report the mean and standard error (across three seeds) of the average absolute error over all target returns. A comparison for each target return is shown in Fig. 7.

| Dataset | Environment | DT | StARformer | DC | RADT (ours) |
|---|---|---|---|---|---|
| medium-replay | ant | $23.2 \pm 1.3$ | $14.1 \pm 3.7$ | $21.5 \pm 3.1$ | $\mathbf{3.5} \pm 0.5$ |
| | halfcheetah | $7.2 \pm 1.8$ | $23.0 \pm 5.0$ | $7.0 \pm 0.8$ | $\mathbf{1.8} \pm 0.6$ |
| | hopper | $12.2 \pm 4.3$ | $14.3 \pm 1.9$ | $5.8 \pm 0.4$ | $\mathbf{4.4} \pm 0.5$ |
| | walker2d | $8.1 \pm 0.3$ | $8.7 \pm 2.4$ | $8.5 \pm 1.6$ | $\mathbf{4.0} \pm 0.8$ |
| medium-expert | ant | $22.0 \pm 2.6$ | $23.1 \pm 0.8$ | $24.9 \pm 1.0$ | $\mathbf{9.2} \pm 0.3$ |
| | halfcheetah | $17.0 \pm 0.7$ | $18.3 \pm 2.3$ | $14.6 \pm 1.6$ | $\mathbf{10.6} \pm 2.0$ |
| | hopper | $12.9 \pm 2.2$ | $12.1 \pm 0.6$ | $8.4 \pm 1.6$ | $\mathbf{4.8} \pm 0.5$ |
| | walker2d | $17.2 \pm 3.7$ | $23.3 \pm 0.3$ | $17.1 \pm 1.7$ | $\mathbf{11.0} \pm 2.3$ |
| medium | ant | $37.3 \pm 2.7$ | $32.8 \pm 1.2$ | $36.8 \pm 3.5$ | $\mathbf{26.3} \pm 3.0$ |
| | halfcheetah | $36.9 \pm 2.8$ | $42.3 \pm 0.6$ | $38.8 \pm 2.0$ | $\mathbf{36.3} \pm 4.4$ |
| | hopper | $16.3 \pm 0.3$ | $14.8 \pm 2.4$ | $13.1 \pm 1.1$ | $\mathbf{6.5} \pm 1.1$ |
| | walker2d | $23.6 \pm 5.7$ | $35.6 \pm 2.5$ | $16.5 \pm 6.4$ | $\mathbf{10.8} \pm 4.1$ |

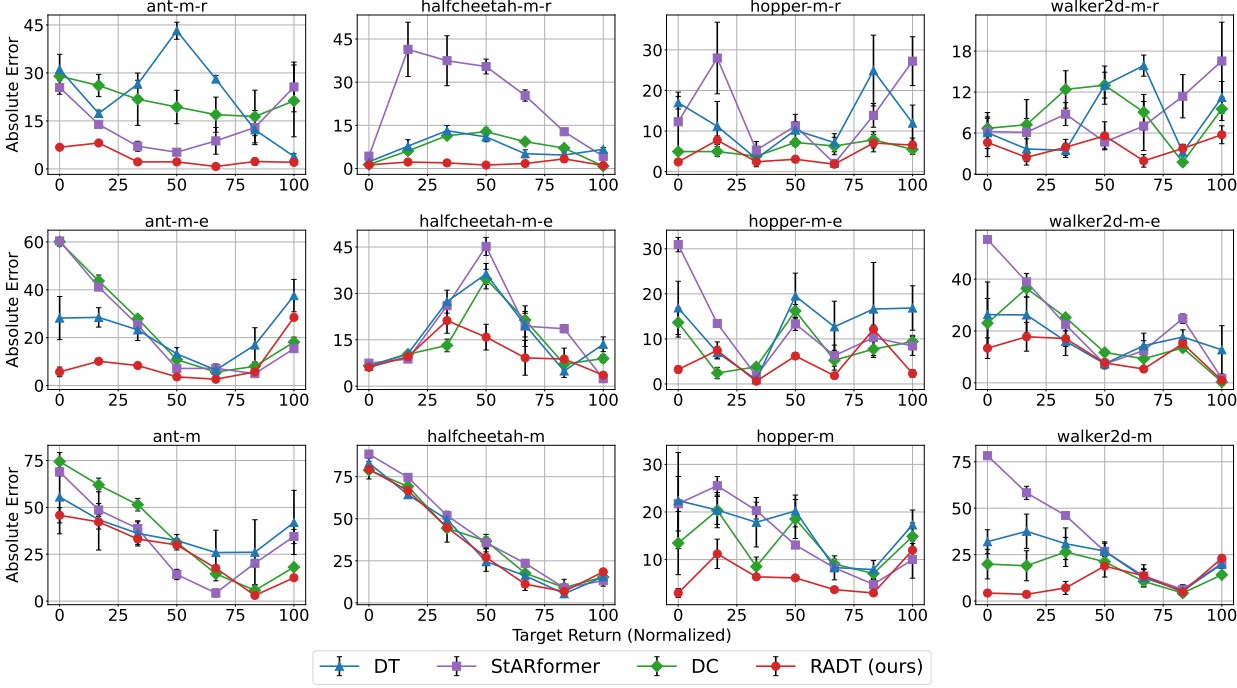

Figure 7: **Absolute error↓ comparison of across target returns in the MuJoCo domain.** We report the mean and standard error over three seeds. The dataset names are shortened: 'medium-replay' to 'm-r', 'medium-expert' to 'm-e', and 'medium' to 'm'.

range of target returns. These results suggest that RADT can effectively align the actual return with the target return in environments requiring fine-grained control with dense rewards.

We observe that differences in the environment affect return alignment. From Fig. 10 in Appendix A.2, we observe that the actual returns are approximately 75 regardless of the target return across all methods. This is likely because the returns of the trajectories in halfcheetah-medium are biased around 75, causing

Table 2: **Absolute error↓ comparison in the Atari domain.** The way to interpret this table is the same as that of Tab. 1. A comparison for each target return is shown in Fig. 8.

| Game | DT | StARformer | DC | RADT (ours) |
|------|-----|-----------|-----|-------------|
| Breakout | $21.8 \pm 5.2$ | $28.8 \pm 8.9$ | $20.1 \pm 3.2$ | $\mathbf{11.9} \pm 3.3$ |
| Pong | $20.4 \pm 0.4$ | $10.9 \pm 3.8$ | $16.6 \pm 3.9$ | $\mathbf{10.4} \pm 2.1$ |
| Qbert | $184.1 \pm 137.0$ | $176.5 \pm 103.6$ | $192.7 \pm 37.2$ | $\mathbf{40.7} \pm 6.3$ |
| Seaquest | $42.5 \pm 18.6$ | $71.0 \pm 4.4$ | $49.6 \pm 22.1$ | $\mathbf{21.0} \pm 4.6$ |

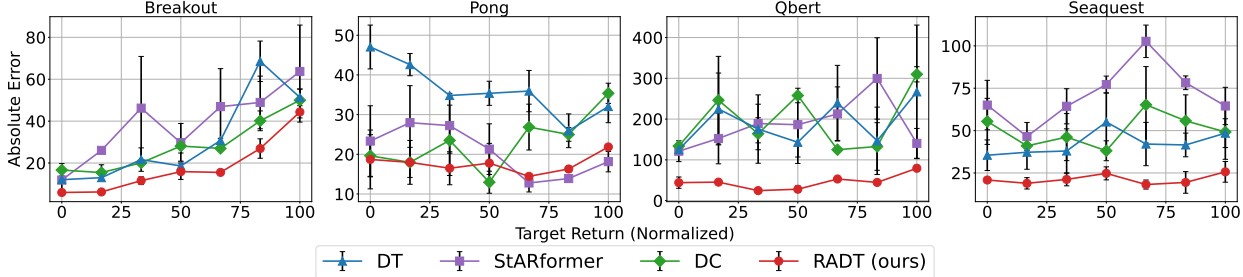

Figure 8: **Absolute error↓ comparison across target returns in the Atari domain.** We report the mean and standard error over three seeds.

the model to overfit to it. Therefore, substantial bias in the dataset's return distribution may reduce the effectiveness of return alignment across most methods.

**Atari Domain.** In the Atari domain, as shown in Tab. 2, RADT outperforms all baseline methods across all tasks. Figure 8 shows that the baselines tend to exhibit large absolute errors for most target returns. This suggests that accurate return modeling in the Atari domain is challenging due to the delays between actions and their rewards. Despite this difficulty, RADT consistently demonstrates small absolute errors across a wide range of target returns. These results indicate that RADT can effectively match actual returns to target returns in environments requiring long-term credit assignment.

However, when examining target returns of 66.7, 83.3, and 100 in Pong, StARformer exhibits the smallest error. We consider this discrepancy to arise from differences in the image encoders used for observations. While RADT uses the same CNN-based image encoder as DT, StARformer employs a more powerful Vision Transformer-based encoder (Dosovitskiy et al., 2021). This suggests that adopting a stronger image encoder could potentially close the performance gap.

## 5.4 Behavior of RADT in Achieving Return Alignment

In Sec 3, we confirmed that DT cannot adjust its actions according to decreases in the return-to-go, leading to an actual return that significantly exceeds the target return. In this section, we compare the behavior of RADT and DT to evaluate whether RADT addresses this issue. For each target return, we perform 100 runs and calculate the mean and standard error of the return-to-go at each step. Figure 2 plots the transitions of return-to-go input to the models during an episode.

For all target returns, DT exhibits a nearly linear decrease in the return-to-go, eventually reaching values far below zero. This indicates that DT does not adjust its behavior in response to decreases in the return-to-go. These findings suggest that DT generates actions largely independent of the return-to-go decrease. In contrast, RADT demonstrates a gradual decrease in return-to-go that converges near zero, significantly reducing the margin by which it exceeds zero compared to DT. This suggests that RADT adaptively adjusts its actions in response to the changing return-to-go.

Table 3: **Absolute error↓ comparison in the ablation study on SeqRA and StepRA.** We conduct the ablation study on the medium-replay dataset in MuJoCo. "w/o X" refers to the ablation of X from RADT. Each value indicates the mean of absolute errors across multiple target returns as defined in Sec. 5.2. We report the mean and standard error over three seeds, and normalize so that the average and standard error of the DT's results are 1.0 respectively.

| Approach | ant | halfcheetah | hopper | walker2d |
|---|---|---|---|---|
| RADT (ours) | $\mathbf{0.15} \pm 0.42$ | $\mathbf{0.25} \pm 0.34$ | $\mathbf{0.36} \pm 0.13$ | $\mathbf{0.50} \pm 2.78$ |
| w/o SeqRA | $0.19 \pm 0.43$ | $0.28 \pm 0.49$ | $0.59 \pm 0.54$ | $0.56 \pm 0.60$ |
| w/o StepRA | $0.42 \pm 2.97$ | $0.32 \pm 0.25$ | $0.93 \pm 0.73$ | $0.63 \pm 3.34$ |
| DT | $1.00 \pm 1.00$ | $1.00 \pm 1.00$ | $1.00 \pm 1.00$ | $1.00 \pm 1.00$ |

Table 4: **Absolute error↓ comparison in the ablation study on SeqRA and StepRA in the Atari domain.** The way to interpret this table is the same as that of Tab 3.

| Approach | Breakout | Pong | Qbert | Seaquest |
|---|---|---|---|---|
| RADT (ours) | $\mathbf{0.55} \pm 0.63$ | $\mathbf{0.51} \pm 4.72$ | $\mathbf{0.22} \pm 0.05$ | $\mathbf{0.49} \pm 0.25$ |
| w/o SeqRA | $0.61 \pm 0.60$ | $0.91 \pm 11.13$ | $0.24 \pm 0.02$ | $0.95 \pm 0.72$ |
| w/o StepRA | $0.57 \pm 0.20$ | $0.59 \pm 7.16$ | $0.24 \pm 0.04$ | $0.81 \pm 1.11$ |
| DT | $1.00 \pm 1.00$ | $1.00 \pm 1.00$ | $1.00 \pm 1.00$ | $1.00 \pm 1.00$ |

## 5.5 Ablation Study

We conduct ablation studies on the two types of return aligners comprising RADT using the medium-replay dataset in MuJoCo and the Atari dataset. The results for the medium-replay dataset in MuJoCo are summarized in Tab. 3, and those for the Atari dataset are presented in Tab. 4. Each value represents the mean of the absolute errors between the actual return and the target return across multiple target returns. These values are then normalized so that the average and standard error of the DT's results are 1.0. The results from both Tab. 3 and Tab. 4 demonstrate that individually introducing either SeqRA or StepRA is effective. Combining both aligners results in the smallest error across all tasks, implying that SeqRA and StepRA effectively complement each other. In the MuJoCo domain, RADT w/o SeqRA outperforms RADT w/o StepRA, indicating that StepRA is particularly advantageous for continuous control tasks with dense rewards in MuJoCo. Conversely, in the Atari domain, RADT w/o StepRA performs equally or better compared to RADT w/o SeqRA, suggesting that SeqRA is beneficial for long-term credit assignment required in the Atari domain.

## 5.6 Comparison on Maximizing Expected Returns

In this section, we evaluate the influence of our proposed method on maximizing expected returns. We use three environments: MuJoCo tasks characterized by dense rewards, AntMaze tasks featuring sparse rewards, and Atari tasks requiring long-term credit assignment. For RADT, we train three instances with different seeds. Each instance is evaluated over 100 episodes for MuJoCo and AntMaze, and 10 episodes for Atari. The resulting average normalized returns are summarized in Tab. 5 for MuJoCo and AntMaze, and in Tab. 7 for Atari. The normalized returns are computed so that 100 represents the score of an expert policy, as per Fu et al. (2020) for MuJoCo and AntMaze, and Hafner et al. (2021) for Atari. Additionally, these tables include the highest episode returns in each dataset.

From Tab. 5 and Tab. 6, RADT achieves performance close to the maximum returns of the datasets in MuJoCo and AntMaze, and its performance is comparable to or exceeds that of DT and DC. Similarly, from Tab. 7, RADT reaches returns near the maximum values of the datasets in Breakout and Pong. In Qbert and Seaquest, although RADT also successfully achieves returns well above the maximum returns of the datasets, it falls short of the baselines specializing in maximizing returns, despite outperforming all approaches in return alignment as shown in Tab. 2. RADT effectively reproduces near-maximum dataset

Table 5: **Performance↑ comparison of return maximization in the MuJoCo domain.** We report the average across three seeds from our simulation results for RADT. The boldface numbers denote the maximum score. We cite the results for DT and DC from their reported scores. We exclude StARformer from the comparison since the original paper does not report results on MuJoCo tasks. The max return of the dataset indicates the highest episode returns for each dataset.

| Dataset | Environment | DC | DT | RADT (ours) | Max Return of Dataset |
|---|---|---|---|---|---|
| medium-replay | halfcheetah | **41.3** | 36.6 | **41.3** ± 0.30 | 42.4 |
| | hopper | 94.2 | 82.7 | **95.7** ± 0.22 | 98.7 |
| | walker2d | 76.6 | 66.6 | 75.9 ± 1.55 | 90.0 |
| medium-expert | halfcheetah | 93.0 | 86.8 | **93.1** ± 0.01 | 92.9 |
| | hopper | **110.4** | 107.6 | **110.4** ± 0.38 | 116.1 |
| | walker2d | 109.6 | 108.1 | **109.7** ± 0.16 | 109.1 |

Table 6: **Performance↑ comparison of return maximization in the AntMaze domain.** We cite the results for DC from their reported scores. We exclude StARformer and DT from the comparison since the original papers do not report results on AntMaze tasks. The way to interpret this table is the same as that of Tab. 5.

| Dataset | DC | RADT (ours) | Max Return of Dataset |
|---|---|---|---|
| umaze | 85.0 | **90.7** ± 4.35 | 100.0 |
| umaze-diverse | 78.5 | **80.7** ± 2.37 | 100.0 |

Table 7: **Performance↑ comparison of return maximization in the Atari domain.** We cite the results for DC, StARformer, and DT from their reported scores. The way to interpret this table is the same as that of Tab. 5.

| Environment | DC | StARformer | DT | RADT (ours) | Max Return of Dataset |
|---|---|---|---|---|---|
| Breakout | 352.7 | **436.1** | 267.5 | 302.6 ± 22.6 | 371.4 |
| Pong | 106.5 | **110.8** | 106.1 | 107.0 ± 2.1 | 116.7 |
| Qbert | **67.0** | 51.2 | 15.4 | 12.8 ± 3.5 | 5.8 |
| Seaquest | **2.7** | 1.7 | 2.5 | 1.2 ± 0.1 | 0.6 |

performance, but faces challenges in scenarios requiring extrapolation to substantially higher target returns beyond the training distribution. We further discuss this limitation in Sec. 7.

## 6 Related work

**Return-conditioned Offline RL** Recent studies have focused on formulating offline reinforcement learning (RL) as a problem of predicting action sequences that are conditioned by goals and rewards (Chen et al., 2021; Janner et al., 2021; Emmons et al., 2022; David et al., 2023; Schmidhuber, 2019; Srivastava et al., 2019). This approach differs from the popular value-based methods (Kumar et al., 2020; Fujimoto & Gu, 2021; Kostrikov et al., 2022) by modeling the relationship between rewards and actions through supervised learning. Decision transformer (DT) (Chen et al., 2021) introduces the concept of desired future returns and improves performance by training the transformer architecture (Vaswani et al., 2017) as a return-conditioned policy. Based on DT, various advancements have been proposed for introducing value functions (Yamagata et al., 2023; Gao et al., 2024; Wang et al., 2024), finetuning models online (Zheng et al., 2022), adjusting

the history length (Wu et al., 2023b), and improving the transformer architecture (Shang et al., 2022; Kim et al., 2024). However, these approaches do not explicitly address the alignment between the actual returns and the target return, which is a central focus of our work.

**Improving Transformer Architecture for Offline RL**   Among the advancements based on DT, we delve deeper into studies that focus on modifying the model architecture similarly to our approach. Some efforts focus on refining the transformer architecture for offline RL. StARformer (Shang et al., 2022) introduces two transformer architectures, one aggregates information at each step, and the other aggregates information across the entire sequence. The image encoding process is improved by dividing the observation images into patches and feeding them into the transformer to enhance step information, similar to Vision Transformer (Dosovitskiy et al., 2021). Decision ConvFormer (Kim et al., 2024) replaces attention with convolution to capture the inherent local dependence pattern of MDP. These methods focus primarily on improving aspects of performance. To the best of our knowledge, this is the first work to explicitly align actual returns with target returns as the main objective in return-conditioned offline RL.

## 7   Discussion and Conclusion

In this paper, we proposed RADT, a novel decision-making model for aligning the actual return with the target return in offline RL. RADT splits the input sequence into return-to-go and state-action sequences, and reflects return-to-go in action generation by uniquely handling the return-to-go sequence. This unique handling includes two strategies that capture long-term dependencies and stepwise relationships within the return-to-go sequence. Experimental results demonstrated that RADT has superior aligning capabilities compared to existing DT-based models. One limitation of our method is a slight increase in computational cost compared to DT. This could potentially be improved by introducing a lightweight attention mechanism, such as Flash Attention (Dao et al., 2022), into our SeqRA. Another limitation is the difficulty in adapting to target returns that significantly exceed the maximum returns within the dataset. This issue could potentially be addressed by enhancing the training data returns through data augmentation techniques Li et al. (2024); Luo et al. (2025). We believe that RADT's alignment capability will improve the usability of offline RL agents in a wide range of applications: creating diverse AI opponents in video games and educational tools, controlling heterogeneous agents in simulations, and efficiently generating human motions for animation or robotics. Investigating how RADT generalizes to multi-agent environments with diverse and adaptive opponents is an exciting avenue for future work.

## Acknowledgments

This work was supported by JST, ACT-X Grant Number JPMJAX23CE, Japan. Kaito Ariu is supported by JSPS KAKENHI Grant No. 23K19986.

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

## Broader Impact

RADT can bring about a positive social impact by enabling adaptation to new application fields such as game production, educational content production, and simulations, as it can control the performance of agents more accurately. However, this advantage also comes with potential negative impacts such as the tracing of user behavior patterns. Such impact can be mitigated by applying methods such as blinding personal information during data generation and collection process.

## A    Additional Experimental Results

To better understand RADT, this section first explains additional analysis of the discrepancies between actual return and target return. Then, we describe an ablation study on adaptive scaling from Sec 4.1, and ablation studies on each sublayer in the Atari domain.

### A.1    Additional Analysis of Discrepancies

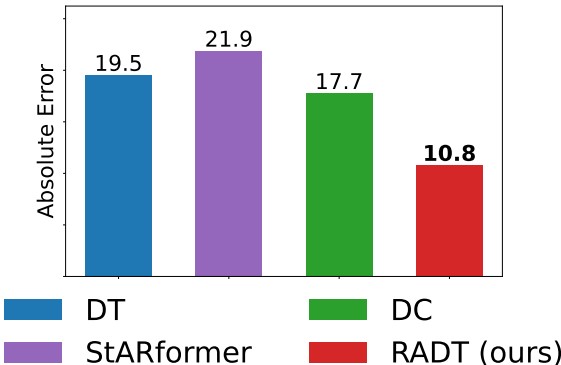

Figure 9: **Absolute errors↓ between the target return and the actual return in the MuJoCo domain.**

We present Fig. 9, which summarizes the discrepancies between the actual return and the target return in the MuJoCo domain. The figure shows the averages of the absolute errors between the actual return and the target return from Fig. 7, across seven target returns and twelve tasks. We normalize these absolute errors by the difference between the top 5% and bottom 5% returns in the dataset [3]. RADT outperforms the baselines, achieving a 44.6% reduction in discrepancies compared to DT. Among the baselines, DC has the smallest discrepancies, while the StARformer has the largest.

To further delve into the results of the MuJoCo domain, we present a comparison of the actual returns in Fig 10. The black dotted line represents $y = x$, indicating that the actual return matches the target return perfectly. The closer to the black dotted line, the better the result. In all tasks except halfcheetah-medium, RADT is closer to the target return than the baseline is. It can be seen that ant-medium and halfcheetah-medium are struggling due to the extremely biased distribution of target returns in the datasets. In some tasks, the baselines show a constant actual return regardless of the input target return (e.g., DT in ant-medium-replay, StARformer in walker2d, DC and StARformer in ant-medium-expert, etc.). We believe this is due to the models overfitting the target return in areas where the data is concentrated.

We present Fig 11, which illustrates the discrepancies between the actual return and the target return in the Atari domain. The values in this figure represent the average absolute errors between the actual return and

---

[3]We normalize returns from each dataset split, unifying the normalized ranges to facilitate comparisons between splits (medium, medium-replay, medium-expert). D4RL's normalization uses the same values across different splits, making it difficult to compare results because the achievable performance limits for trained agents vary significantly between splits.

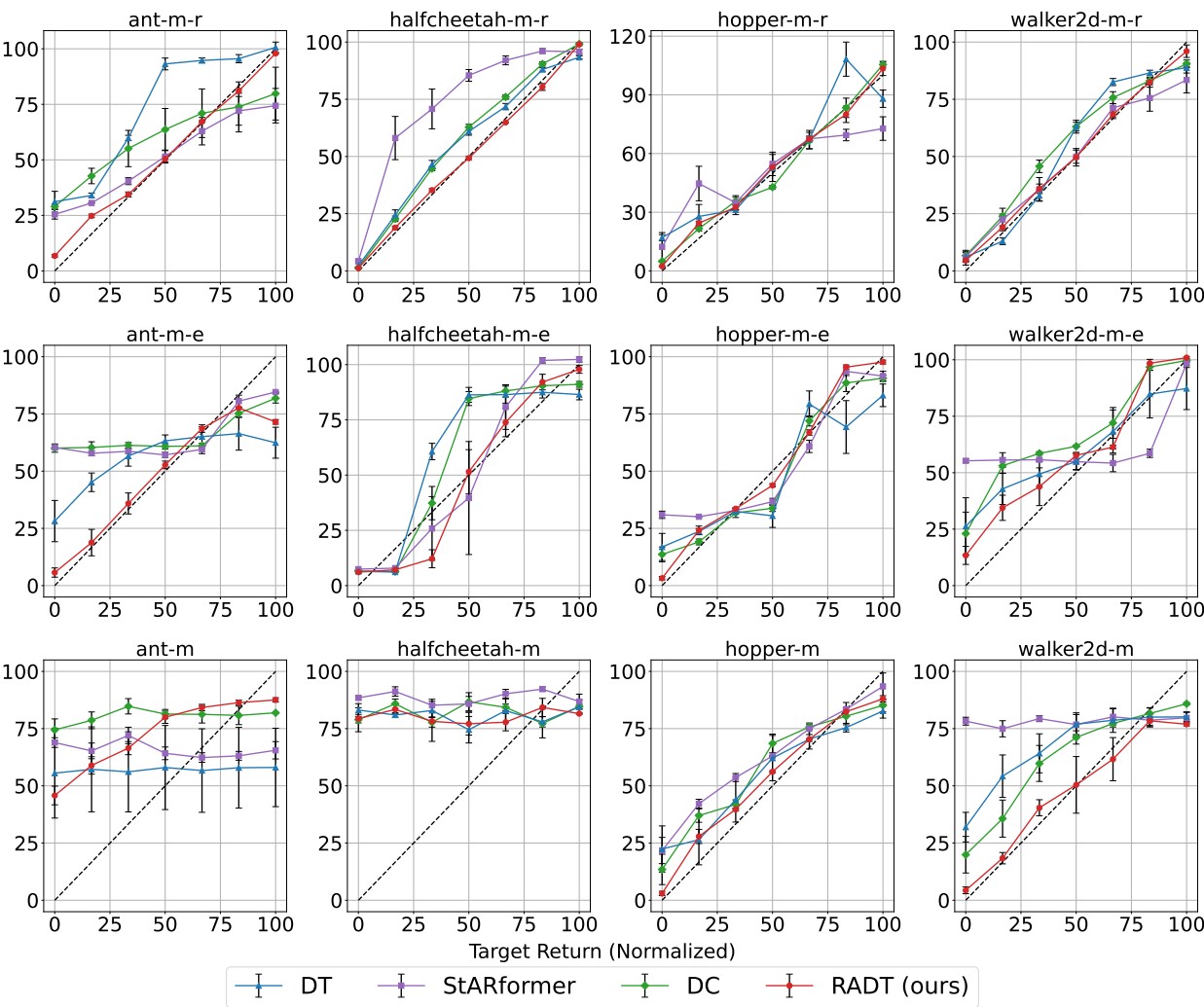

Figure 10: **Comparisons of actual returns per target return in the experiments of Fig. 7 for MuJoCo.** Each column represents a task. The x-axis represents the target return, and the y-axis represents the actual return. The x-axis and y-axis are normalized in the same way as in Fig. 7. Target returns are set in the same way as Fig. 7. The black dotted line represents $y = x$, indicating that the actual return matches the target return perfectly. We report the mean and standard error over three seeds. The dataset names are shortened:'medium-replay' to 'm-r', 'medium-expert' to 'm-e', and 'medium' to 'm'.

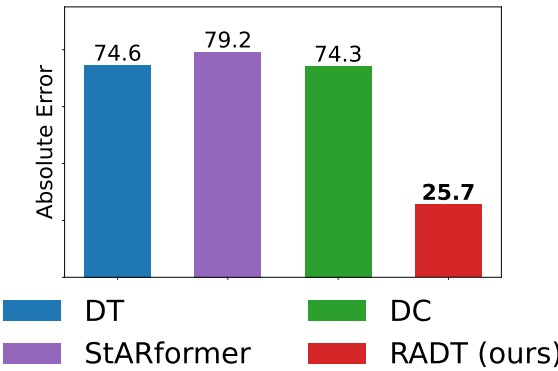

Figure 11: **Absolute errors↓ between the target return and the actual return in the Atari domain.**

the target return, averaged over seven target returns and four tasks. These absolute errors are normalized by the difference between the top 5% and the bottom 5% of returns in the dataset. RADT significantly outperforms the baseline methods. Specifically, it achieves a 65.5% reduction in discrepancies compared to DT, which is a greater reduction margin than what we observed in the MuJoCo domain. Among the baseline methods, DC exhibits the smallest discrepancies, while StARformer shows the largest, a pattern that is consistent with the MuJoCo domain.

## A.2   Additional Ablation Study

We conduct an ablation study on the adaptive scaling in SeqRA on the medium-replay dataset in MuJoCo and the Atari dataset. The experimental setup is the same as described in Sec.5.5. The results are summarized in Tab. 8 and Tab. 9. These results demonstrate that introducing adaptive scaling (RADT w/o StepRA) outperforms the case where only the attention mechanism of SeqRA is used (RADT w/o StepRA and Adaptive Scaling). This suggests that adaptive scaling effectively merges $z$, which includes information from the return sequence, with the state-action sequence $\tau_{sa}$.

Table 8: **Absolute error↓ comparison in the ablation study for the adaptive scaling in SeqRA on the medium-replay dataset in MuJoCo domain.** The way to interpret this table is the same as that of Tab 3.

| Approach | ant | halfcheetah | hopper | walker2d |
|---|---|---|---|---|
| RADT (ours) | $\mathbf{0.15} \pm 0.42$ | $\mathbf{0.25} \pm 0.34$ | $\mathbf{0.36} \pm 0.24$ | $\mathbf{0.50} \pm 2.78$ |
| w/o StepRA | $0.42 \pm 2.97$ | $0.32 \pm 0.25$ | $0.93 \pm 0.73$ | $0.63 \pm 3.34$ |
| w/o StepRA and Adaptive Scaling | $0.74 \pm 2.94$ | $0.85 \pm 0.45$ | $0.99 \pm 0.73$ | $0.87 \pm 2.00$ |
| DT | $1.00 \pm 1.00$ | $1.00 \pm 1.00$ | $1.00 \pm 1.00$ | $1.00 \pm 1.00$ |

Table 9: **Absolute error↓ comparison in the ablation study for the adaptive scaling in SeqRA on the Atari domain.** The way to interpret this table is the same as that of Tab 3.

| Approach | Breakout | Pong | Qbert | Seaquest |
|---|---|---|---|---|
| RADT (ours) | $\mathbf{0.55} \pm 0.63$ | $\mathbf{0.51} \pm 4.72$ | $\mathbf{0.22} \pm 0.05$ | $\mathbf{0.49} \pm 0.25$ |
| w/o StepRA | $0.57 \pm 0.20$ | $0.59 \pm 7.16$ | $0.24 \pm 0.04$ | $0.81 \pm 1.11$ |
| w/o StepRA and Adaptive Scaling | $0.72 \pm 1.52$ | $0.86 \pm 2.16$ | $0.26 \pm 0.07$ | $0.86 \pm 1.22$ |
| DT | $1.00 \pm 1.00$ | $1.00 \pm 1.00$ | $1.00 \pm 1.00$ | $1.00 \pm 1.00$ |

### A.3 Additional Analysis of Parameter Size

We compare RADT and models where we increase the number of blocks in DT to bring the number of parameters closer to RADT. Table 10 shows the parameters for each model on the medium-replay dataset of MuJoCo. Because RADT adds both SeqRA and StepRA to the transformer block, it has 1.6 times the number of parameters compared to DT. By increasing the number of blocks in DT to six, it surpasses the number of parameters of RADT. Therefore, we compare RADT with a DT that has six blocks and present the results on the medium-replay dataset of MuJoCo in Tab. RADT outperforms the DT with six blocks. On the other hand, we find that DT with 6 blocks performs worse than that with 3 blocks in the hopper and walker2d environments. This result suggests that simply increasing the number of parameters of DT does not improve return alignment, and that the improved model structure of RADT is effective.

Table 10: **Comparison of parameters.** Since the dimensions of the state and action spaces vary for each dataset, the model size differs depending on the dataset.

| Environment | DT(3 blocks) | DT(5 blocks) | DT(6 blocks) | RADT(3 blocks) |
|---|---|---|---|---|
| ant | 740,232 | 1,136,776 | 1,335,048 | 1,219,848 |
| halfcheetah | 727,686 | 1,124,230 | 1,322,502 | 1,207,302 |
| hopper | 726,147 | 1,122,691 | 1,320,963 | 1,205,763 |
| walker2d | 727,686 | 1,124,230 | 1,322,502 | 1,207,302 |

Table 11: **Absolute error$\downarrow$ comparison between RADT and DT with increased parameters.** We use the medium-replay dataset in MuJoCo. The way to interpret this table is the same as that of Tab 1.

| Environment | DT(3 blocks) | DT(6 blocks) | RADT(3 blocks) |
|---|---|---|---|
| ant | $23.2 \pm 1.3$ | $16.8 \pm 4.5$ | $\mathbf{3.5} \pm 0.5$ |
| halfcheetah | $7.2 \pm 1.8$ | $5.5 \pm 0.5$ | $\mathbf{1.8} \pm 0.6$ |
| hopper | $12.2 \pm 4.3$ | $12.8 \pm 2.4$ | $\mathbf{4.4} \pm 0.5$ |
| walker2d | $8.1 \pm 0.3$ | $10.0 \pm 0.7$ | $\mathbf{4.0} \pm 0.8$ |

## B   Baseline Details

We use the model code for DT, StARformer, and DC from the following sources. DT: `https://github.com/kzl/decision-transformer`. StARformer: `https://github.com/elicassion/StARformer`. DC: `https://openreview.net/forum?id=af2c8EaKl8`. Although StARformer uses step-by-step rewards instead of returns, in our experiments, we employ return-conditioning using returns. This modification allows StARformer to condition action generation on target return. The original paper (Shang et al., 2022) states that this modification has a minimal impact on performance. For visual observations in the Atari domain, RADT and DC use the same CNN encoder as DT. StARformer, in addition to the CNN encoder, also incorporates more powerful patch-based embeddings like Vision Transformer (Dosovitskiy et al., 2021).

The baseline results for aligning the actual return with the target return (Sec. 5.3) and the ablation study (Sec. 5.5) are from our simulations. The hyperparameters for each method in our simulations are set according to the defaults specified in their original papers or open-source codebases. The baseline results for maximizing the expected return (Sec. 5.6) stem from the original papers or third-party reproductions.

## C   Experimental Details

For the reproducibility of the experiments, this section explains the comparison of computational costs and the hyperparameters used in the experiments.

Table 12: **Comparision of computational cost.**

| Method | Training Time (s) | GPU memory usage (GiB) |
|---|---|---|
| DT | 363 | 0.030 |
| RADT | 466 | 0.034 |

## C.1 Comparison of Computational Cost

Table 12 shows a comparison of the computational costs of DT and RADT. We compare the training time and GPU memory usage incurred when running $10^4$ iterations of training on the hopper-medium-replay dataset. In this comparison, we use an NVIDIA A100 GPU. RADT has slight increases in computation time and memory usage from DT. We believe these increases are due to the addition of SeqRA and StepRA. The computational costs of RADT could potentially be improved by introducing efficient attention mechanisms such as Flash Attention (Dao et al., 2022).

## C.2 Hyperparameters

Our implementation of RADT is based on the public codebase of DT. We used an Nvidia A100 GPU for training in the Atari and MuJoCo domains. The full list of hyperparameters of RADT is found in Tab. 13 and Tab. 14. The hyperparameter settings of RADT are the same in both aligning and maximizing.

Table 13: **Hyperparameters settings of RADT in the MuJoCo domain and the AntMaze domain.** The dataset names are shortened: 'medium' to 'm', 'medium-expert' to 'm-e', 'umaze' to 'u', and 'umaze-diverse' to 'u-d'.

| Hyperparameter | Value |
|---|---|
| Number of blocks | 2, hopper-m |
| | 3, otherwise |
| Number of heads | 1 |
| Embedding dimension | 256, ant-m-e, halfcheetah-m, antmaze-u, antmaze-u-d |
| | 128, otherwise |
| Batch size | 256, antmaze-u, antmaze-u-d |
| | 64, otherwise |
| Nonlinearity function | GELU, transformer |
| | SiLU, StepRA |
| Context length K | 20 |
| Dropout | 0.1 |
| Learning rate | $10^{-4}$ |
| Grad norm clip | 0.25 |
| Weight decay | $10^{-4}$ |
| Learning rate decay | Linear warmup for first $10^4$ steps |

# D Additional Real-World Application Example

We describe the additional real-world application that requires return alignment, as mentioned in Sec. 1.

**Example 3** (Agent-based modeling and simulation)**.** Agent-based modeling and simulation (Macal & North, 2005) involves examining system dynamics by representing them as collections of interacting, autonomous agents. In this context, incorporating heterogeneous behaviors (Hu et al., 2023) and controllable poli-

Table 14: **Hyperparameters settings of RADT in the Atari domain.**

| Hyperparameter | Value |
| --- | --- |
| Number of blocks | 6 |
| Number of heads | 8 |
| Embedding dimension | 128 |
| Batch size | 512 Pong |
| | 128 Breakout, Qbert, Seaquest |
| Context length K | 50 Pong |
| | 30 Breakout, Qbert, Seaquest |
| Nonlinearity | ReLU encoder |
| | GELU transformer |
| | SiLU StepRA |
| Encoder channels | $32, 64, 64$ |
| Encoder filter size | $8 \times 8, 4 \times 4, 3 \times 3$ |
| Encoder strides | $4, 2, 1$ |
| Max epochs | 15 |
| Dropout | 0.1 |
| Learning rate | $6 \times 10^{-4}$ |
| Adam betas | $(0.9, 0.95)$ |
| Grad norm clip | 1.0 |
| Weight decay | 0.1 |
| Learning rate decay | Linear warmup and cosine decay |
| Warmup tokens | $512 * 20$ |
| Final tokens | $2 * 500000 * K$ |

cies (Panayiotou et al., 2022) is essential. Our approach introduces a controlled variety of agent behaviors by adjusting their performance levels based on a specified target return.

