# OpenReview forum: "Return-Aligned Decision Transformer"
_TMLR — Accepted by TMLR_

### Review · Reviewer_i1BT · 2025-02-04

**Summary Of Contributions:**

This paper proposes a few modifications to the decision transformer architecture in the context of return-conditioned reinforcement learning such that rewards achieved by the agent aligns more closely with the rewards it is conditioned on. The main modifications are "sequence return aligner" which introduces additive interactions with state-action tokens and "stepwise return aligner" which introduces multiplicative interactions with state-action tokens. The experiments show that the proposed approach significantly improves return alignment accuracy in Mujoco and Atari environments compared to baselines (including the vanilla decision transformer).

**Audience:**

Yes

**Broader Impact Concerns:**

There's no broader impact concerns.

**Claims And Evidence:**

Yes

**Requested Changes:**

I don't have other requested changes.

**Strengths And Weaknesses:**

**Strength**
* The problem is well-motivated and related to the subject of controllable agents.
* The finding of the attention mechanism in the vanilla decision transformer being a possible main contributor to misaligned returns is a good one.

**Weakness/questions**
* The motivations for the proposed architecture modifications were not super clear. It seemed that the only goal was to ensure reward-related information is routed to the output head.
* Intuitively in RL via SL, data quality/diversity is important. We would expect models trained on data with diverse returns to be better return-aligned. Why do we see medium-replay having better overall alignment? Is that because it has more diverse returns?
* Why in certain environments the return error can be as high as 75 (e.g., halfcheetah-m)? I think it'd be good to comment on the effect of the environments on the results, e.g. the difficulty of achieving a specific level of return is not necessarily in every environment.

---

> ### Author Response · Authors · 2025-04-18
> **Official Comment by Authors**
>
> We are grateful for your thorough review and insightful recommendations. Your feedback will undoubtedly help us strengthen the revised manuscript.
>
> > The motivations for the proposed architecture modifications were not super clear. It seemed that the only goal was to ensure reward-related information is routed to the output head.
> >
>
> The proposed architecture aims to consistently align with the return-to-go throughout the entire model, rather than simply routing reward-related information to the final output head. Specifically, SeqRA and StepRA embed the return-to-go token features into the state-action sequence, which allows the feed-forward networks and self-attention modules in subsequent blocks to continually refine the representations based on the return-to-go. RADT gradually guides the features of the state-action sequence to align with the return-to-go as each block is passed through.
>
> > Intuitively in RL via SL, data quality/diversity is important. We would expect models trained on data with diverse returns to be better return-aligned. Why do we see medium-replay having better overall alignment? Is that because it has more diverse returns?
> >
>
> As you mentioned, the reason why medium-replay performs better than other splits is likely because medium-replay includes trajectories with more diverse returns. The medium-replay split has trajectories extracted from the replay buffer of the medium policy, resulting in a wide range of returns up to the performance level of the medium policy. In contrast, the returns in the medium split are concentrated around the performance of the medium policy, while the returns in the medium-expert split are concentrated around the performances of the medium or expert policies.
>
> > Why in certain environments the return error can be as high as 75 (e.g., halfcheetah-m)? I think it'd be good to comment on the effect of the environments on the results, e.g. the difficulty of achieving a specific level of return is not necessarily in every environment.
> >
>
> We have added the following discussion to the MuJoCo Domain in Section 5.3.
>
> *We observe that differences in the environment affect return alignment. From Fig.10 in Appendix A.2, we observe that the actual returns are approximately 75 regardless of the target return across all methods. This is likely because the returns of the trajectories in halfcheetah-medium are biased around 75, causing the model to overfit to it. Therefore, substantial bias in the dataset’s return distribution may reduce the effectiveness of return alignment across most methods.*

---

> > ### Comment · Reviewer_i1BT · 2025-04-18
> >
> > Thank the authors for the clarification. This is a nice empirical paper and I have no more questions.

---

### Review · Reviewer_TojF · 2025-02-08

**Summary Of Contributions:**

This paper proposes a modification to the architecture of a decisiton transformer (DT) so as to put more focus on the rewards/returns at each layer called RADT. By doing so, the model improves in terms of the error of the desired return compared to the actual return. Experiments verify this on mujoco and atari domains.

**Audience:**

Yes

**Claims And Evidence:**

Yes

**Requested Changes:**

Overall, I think this is a very solid paper with a clear contribution and relatively thorough experimentation. I would just like to see the one extra experiment listed above for completeness.

**Strengths And Weaknesses:**

## Strengths

1. The paper proposes a simple and intuitive architectural change to force the model to incorporate reward information more directly.

2. Experiments verify that indeed the DT model is often ignoring the conditioned returns and thus overshooting the desired return.

3. Experiments clearly show large improvements in terms of the calibration of RADT in the sense that the actual returns are much closer to the desired conditioned returns across environments.

4. Ablation experiments illustrate that both sequence-level and step-level conditioning are useful components of RADT.

5. The presentation is generally clear.

## Weaknesses

1. One issue is that the proposed architecture changes add parameters within every block. As a result the RADT architecture is larger than the DT baselines. This is mentioned briefly in the appendix discussing computational time and memory requirements. However, it would seem to be important to add a parameter-matched baseline DT with slightly larger model so that it matches the parameters in the RADT model.

---

> ### Author Response · Authors · 2025-04-18
> **Official Comment by Authors**
>
> Thank you very much for your thoughtful and constructive feedback. Your comments have been instrumental in improving our paper.
>
> > One issue is that the proposed architecture changes add parameters within every block. As a result the RADT architecture is larger than the DT baselines. This is mentioned briefly in the appendix discussing computational time and memory requirements. However, it would seem to be important to add a parameter-matched baseline DT with slightly larger model so that it matches the parameters in the RADT model.
> >
>
> We have added a comparison between the parameter-matched baseline DT and RADT in Appendix A.3.

---

### Review · Reviewer_2ThK · 2025-04-03

**Summary Of Contributions:**

This paper proposes a new attention model for offline RL, treating offline RL as a sequential modelling problem. Compared to prior work, the model gives higher attention to the return-to-go, effectively aligning the actual return with the target return-to-go. Additionally, it achieves better performance in many MuJoCo environments.

**Audience:**

Yes

**Claims And Evidence:**

Yes

**Requested Changes:**

1. In the second paragraph of Section 3, the paper states that “when the majority of the attention scores are allocated to state or action tokens, the allocation to return-to-go tokens decreases.” It would be helpful if the authors could further explain why the attention scores tend to concentrate on state or action tokens. Why does the return-to-go token receive inadequate attention in this setting?

2. I would suggest that the authors improve the completeness of the experiments. Since the paper’s claim is direct and does not require overly complex analysis, it would be helpful to compare the performance results (Section 5.6) and conduct the ablation study (Section 5.5) in both the MuJoCo and Atari domains, using a wider range of baselines.

3. The authors should improve Section 6 by better reflecting current research methods and emphasising how their approach differs. Are there other methods that modify the model architecture to improve aspects of performance in modelling sequence RL? Is this the first paper to explicitly align predicted returns with return-to-go? Rather than simply listing related work, it would be more informative to provide a critical comparison and highlight the unique contributions of this paper.

4. I don’t think Figure 1 is necessary. The text description is clear enough.

**Strengths And Weaknesses:**

**Strength**

1. When I first started reading this paper, I found it difficult to think of a scenario where a suboptimal policy would be needed. However, the three examples addressed all my concerns. The authors also clearly explain the limitations of DT and motivate their idea effectively.

2. The paper’s main claim is clear and well-supported by the experimental results.

**Weaknesses**

No clear weaknesses.

---

> ### Author Response · Authors · 2025-04-18
> **Official Comment by Authors**
>
> Your detailed and critical suggestions are greatly appreciated. We have carefully considered your feedback and found it extremely valuable in refining our paper.
>
> > 1. In the second paragraph of Section 3, the paper states that “when the majority of the attention scores are allocated to state or action tokens, the allocation to return-to-go tokens decreases.” It would be helpful if the authors could further explain why the attention scores tend to concentrate on state or action tokens. Why does the return-to-go token receive inadequate attention in this setting?
> >
>
> We have added the following content at the end of Section 3.
>
> *We believe this tendency arises because the training data includes samples from Markov policies that do not take return as input. In such training data, return-to-go tokens may be less informative for action prediction than state and action tokens. As a result, DT pays insufficient attention to return-to-go tokens, instead disproportionately focusing on state and action tokens.*
>
> > 2. I would suggest that the authors improve the completeness of the experiments. Since the paper’s claim is direct and does not require overly complex analysis, it would be helpful to compare the performance results (Section 5.6) and conduct the ablation study (Section 5.5) in both the MuJoCo and Atari domains, using a wider range of baselines.
> >
>
> We have made the following revisions to the paper:
>
> - Included performance results for the Atari domain in Section 5.6.
> - Relocated the Atari domain ablation study results from Table 5 in Appendix A.2 to Section 5.5.
>
> > 3. The authors should improve Section 6 by better reflecting current research methods and emphasising how their approach differs. Are there other methods that modify the model architecture to improve aspects of performance in modelling sequence RL? Is this the first paper to explicitly align predicted returns with return-to-go? Rather than simply listing related work, it would be more informative to provide a critical comparison and highlight the unique contributions of this paper.
> >
>
> We have revised Section 6 to make the differences between our method and other studies clearer.
>
> > 4. I don’t think Figure 1 is necessary. The text description is clear enough.
> >
>
> We have deleted the figure titled “Performance-Controllable Model”.

---

> > ### Comment · Reviewer_2ThK · 2025-04-20
> >
> > Thank you for your reply. Most of my concerns have been addressed. However, I still have questions regarding Q1. I agree that the training data are collected from a policy that does not explicitly condition on the return-to-go. However, I’m not convinced that this necessarily implies that the return-to-go tokens are less informative for action prediction. Could you clarify why the lack of explicit conditioning would reduce the attention on the return-to-go tokens in this context?

---

> > > ### Author Response · Authors · 2025-04-21
> > > **Official Comment by Authors**
> > >
> > > Thank you for your insightful comment.
> > > We agree with your observation that return-to-go is not inherently less informative for predicting actions. Our point is that the next action in the training data depends on the state because the policy used during data collection is Markovian and thus conditions only on the current state. As a result, there is a stronger correlation between states and next actions than between returns-to-go and next actions. When the DT model is trained on such data, it tends to allocate most of its attention to the state tokens. Consequently, attention to return-to-go tokens is relatively decreased, as illustrated on the left side of Figure 1.

---

### Review · Reviewer_914B · 2025-04-09

**Summary Of Contributions:**

The paper proposed an adapted transformer architecture for decision transformers such that improves the capability of exactly achieving target returns instead of higher returns. For that, two aligners are proposed. The new architecture is evaluated on Atari and Mujoco benchmarks for multiple trajectory data sets. The evaluation shows a consistent performance improvement compared to other state-of-the-art approaches on the selected benchmarks.

**Audience:**

Yes

**Claims And Evidence:**

Yes

**Requested Changes:**

- Example 2 and 3 in Sec. 1 are very similar and I would suggest removing one of them
- Fig. 2 was not easy to interpret for me at that point of the paper and it is unclear from which experiments the values stem. I would suggest making the figure more stylistic, making the caption clearer on where these numbers stem from, or removing the figure
- Sec. 3 could be significantly shortened, so that it briefly states the hypothesis and explains the figure.
- I would have expected that low target returns are easier to achieve than high, but it looks like that if there is a difference within a benchmark for the RADT performance, it is mainly for lower returns. Do you have an intuition for that?
- For the game example from the introduction, the performance of an RADT agent playing a human player depends on their interaction. Two beginner players could be lacking in two very different ways, so the RADT agent is only a good competitor for one of them. The benchmarks you consider are single-agent benchmarks. Do you think your approach generalizes to multi-agent setups well? What could be good first benchmarks to investigate such setups?

**Strengths And Weaknesses:**

Strengths:
- The paper is clearly motivated (especially Fig 3 showcases the addressed problem well)
- The aligner modules are intuitive
- The evaluation is solid since it compares multiple state-of-the-art approaches on common benchmarks and shows the improvement of achieving specified target returns of the RADT

Weaknesses:
- The motivation and Sec. 3 could be shortened
- Limitations and the differences between different benchmarks should be discussed in more detail

---

> ### Author Response · Authors · 2025-04-18
> **Official Comment by Authors**
>
> We sincerely appreciate your insightful comments and valuable suggestions, which have significantly helped us enhance the quality and clarity of our manuscript.
>
> > The motivation and Sec. 3 could be shortened
> >
>
> > Sec. 3 could be significantly shortened, so that it briefly states the hypothesis and explains the figure.
> >
>
> We have revised Section 3 to explain the hypotheses and figures more concisely.
>
> > Limitations and the differences between different benchmarks should be discussed in more detail
> >
>
> We have added discussions regarding the differences between benchmarks in Sections 5.3, 5.5, and 5.6. Additionally, we have expanded the discussion on limitations in the conclusion.
>
> > Example 2 and 3 in Sec. 1 are very similar and I would suggest removing one of them
> >
>
> We have kept Example 3 in Section 1 and moved Example 2 to the Appendix D.
>
> > Fig. 2 was not easy to interpret for me at that point of the paper and it is unclear from which experiments the values stem. I would suggest making the figure more stylistic, making the caption clearer on where these numbers stem from, or removing the figure
> >
>
> We have clarified the sources of the numbers in the captions and revised the style of the figure.
>
> > I would have expected that low target returns are easier to achieve than high, but it looks like that if there is a difference within a benchmark for the RADT performance, it is mainly for lower returns. Do you have an intuition for that?
> >
>
> In some benchmarks, such as ant-medium-expert and walker2d-medium, the significant difference between RADT and the baselines at low target returns is due to the dataset of those benchmarks containing very few low-return trajectories. Trajectories whose returns fall below 50 (as shown in Figure 7) make up 30.0 % of the total in ant‑medium‑expert and 20.7 % in walker2d‑medium. The baseline models a weak dependency on the target return and tends to have fixed performance in specific return regions. In datasets with many high-return trajectories, the model is likely to be biased towards higher returns, resulting in larger errors in scenarios with low target returns. In contrast, RADT explicitly incorporates dependency on the target return, allowing it to adaptively adjust its behavior even when a low target return is specified. As a result, RADT significantly improves consistency with the target return and greatly reduces errors in these low target return settings.
>
> We have added this explanation to Section 5.3.
>
> > For the game example from the introduction, the performance of an RADT agent playing a human player depends on their interaction. Two beginner players could be lacking in two very different ways, so the RADT agent is only a good competitor for one of them. The benchmarks you consider are single-agent benchmarks. Do you think your approach generalizes to multi-agent setups well? What could be good first benchmarks to investigate such setups?
> >
>
> As you pointed out, the performance of an RADT agent in competitive games depends heavily on interactions with the human player. If diverse opponent behaviors are included in the training data, RADT could generalize well to multi-agent setups by flexibly adapting its strategy. However, in this study, our primary focus was validating that RADT can effectively align with a given target return, leading us initially to select single-agent benchmarks.
>
> For initial exploration into multi-agent environments, benchmarks such as "Othello (Reversi)" or "Connect Four" would be ideal due to their simple yet strategic rules, manageable state spaces, and diverse opponent behaviors. We believe this is an exciting direction for future work. Additionally, we have added this content to the conclusion.

---

### Decision · Action_Editor_GvgW · 2025-04-28

**Recommendation:** Accept with minor revision

**Comment:**

All reviewers recommended acceptance. There is a consensus that this paper identifies a meaningful research question, proposes a well-designed solution, and presents sufficient empirical evidence to demonstrate its effectiveness. It is a solid empirical contribution, aligning well with TMLR’s emphasis on technical soundness over subjective significance or novelty. As stated earlier, some minor points deserve improvement, hence the decision to *accept with minor revision*.

**Audience:**

Transformer-based architecture for RL is of broad interest to the in TMLR's audience.

**Claims And Evidence:**

The claims made in the submission are generally supported by accurate, convincing and clear evidence. Some minor changes have been requested by reviewers to improve clarity on the advantages and drawbacks of the approach. and more globally on the paper.

---

> ### Author Response · Authors · 2025-06-06
> **Official Comment by Authors**
>
> Dear Action Editor,
>
> Thank you very much for your encouraging feedback and for recommending acceptance of our paper with minor revisions. We also extend our sincere gratitude to all reviewers for their thorough evaluations and insightful comments, which greatly assisted us in enhancing the manuscript.
>
> We have carefully addressed all concerns and suggestions in our revised manuscript, and we have submitted the camera-ready version accordingly. Additionally, we have provided a public GitHub link to promote transparency and facilitate reproducibility.
>
> Thank you again for your valuable assistance throughout the review process.